# Nitrate-Alkaline Pulp from Non-Wood Plants

**DOI:** 10.3390/ma14133673

**Published:** 2021-07-01

**Authors:** Kateřina Hájková, Jiří Bouček, Petr Procházka, Petr Kalous, Dominik Budský

**Affiliations:** 1Department of Wood Processing and Biomaterials, Faculty of Forestry and Wood Sciences, Czech University of Life Science Prague, Kamýcká 1176, 165 21 Prague, Czech Republic; 2Department of Applied Ecology, Faculty of Environmental Sciences, Czech University of Life Science Prague, Kamýcká 129, 165 21 Prague, Czech Republic; 3Department of Economics, Faculty of Economics and Administration, Czech University of Life Sciences Prague, Kamýcká 129, 165 00 Prague, Czech Republic; pprochazka@pef.czu.cz; 4Secondary and Higher Vocational School of Packaging Technology in Štětí, Kostelní 134, 411 08 Štětí, Czech Republic; kalous@odbornaskola.cz (P.K.); budsky@odbornaskola.cz (D.B.)

**Keywords:** nitrate-alkaline pulp, black mustard, camelina, tensile index, chemical analysis

## Abstract

Because there is a lack of wood resources in many countries, this work focused on pulp and paper production from the waste and agricultural residues of non-wood plants. The work aimed to pulp the nitrate-alkaline of black mustard (*Brassica Nigra* L.) and camelina (*Camelina Sativa* L.). The black mustard and the camelina were selected due to the expanding planted areas of these crops in the Czech Republic. To characterize the chemical composition of black mustard and camelina, cellulose, lignin, ash, and extractives were determined. Raw alpha-cellulose, beta-cellulose, and gamma-cellulose were also measured. The results showed that the content of lignin in non-wood plants is lower than that in softwoods. The cooked pulp was characterized by the delignification degree–Kappa number. Additionally, handsheet papers were made for selected samples of pulp. The handsheet papers were characterized by tensile index, breaking length, and smoothness and compared with commonly available papers.

## 1. Introduction

Besides wood pulp, the pulp and paper industry uses fibers and agricultural waste. The fibers of linen, hemp, and cotton have been processed since the beginnings of papermaking. These fibers are used for special types of paper. Even straw, sugarcane, cotton, and other non-wood plants are used in the cellulose and paper industry. In the case of choosing the raw material, it is important to know the chemical, morphological composition, and structure of the plants [1].

Wood is the primary source for the paper industry nowadays. There is a lack of wood sources which suits the growing inquiry in a lot of countries. Besides wood pulp, this causes the fibers and waste from the agricultural industry to be used for pulp and paper making. Seventy percent of raw materials in the pulp and paper industry in China and India come from non-wood plants [2].

Hemp, sisal, and linen belong among the plants that are used in papermaking. The other raw material, which is added to banknote paper production, is cotton. The field planting areas in the Czech Republic have significantly changed during the past few years, according to the planted crops. The cereals sown on the largest planting areas are, for example, wheat, barley, or rapeseed, but there has been significant growth of corn, poppy, mustard, and camelina planting areas recently. Figure 1 shows that there is a significant increase in the share of technical crops together with rapeseed (canola) over the last twenty years. This is also true for the area of cropland devoted to poppy seed. A decline in barley, rye, and potatoes means it grows fewer food crops in the Czech Republic over the observed time.

The non-wood plants have chemical and morphological characteristics different from the wood materials, and this is the reason for other cooking and bleaching conditions [4].

The non-wood plants are processed by the chemical method, specifically the amaranth, atriplex and Jerusalem artichoke [5], and rapeseed straw [6,7]. Wheat, rapeseed, amaranth, and lavatera are cooked in peracetic acid [8], and rapeseed straw produced the neutral sulfite semi-chemical pulp [9,10].

Another chemical method for pulp making can be the nitrate-alkaline process, where during cooking in the acid, the cellulose–lignin bond is released hydrolytically. The lignin is nitrated and partly oxidized into nitro-lignin, which is soluble in alkalies [11].

This paper aimed to cook the nitrate-alkaline pulp of black mustard (*Brassica Nigra* L.) and camelina (*Camelina Sativa* L.) and to examine the mechanical properties of paper made from these pulps. We focused this study on examining the chemical pulping of agricultural crops and widening the results by the other potential raw materials for cellulose fibers.

## 2. Materials and Methods

On the fields in the highlands of the Ore Mountains, the black mustard (*Brassica Nigra* L.) is grown, whereas, in the lowlands of the Podyjí area, the camelina (*Camelina Sativa* L.) is harvested. The raw materials comprise stems and fruits, which make up to a quarter of the whole quantity. The chemical analysis was performed according to the TAPPI Test methods before the cooking [12]. The analysis for randomly taken samples were determined in a laboratory vibratory mill, specifically ash (TAPPI T 211om-02) and silica (TAPPI T 245 cm-98) contents, as well as extractive contents by Soxhlet extraction with acetone, ethanol, and with a 7:3 mixture of ethanol to toluene by volume (TAPPI T 280 pm-99), as well as Klason lignin (TAPPI T 22 cm-02) using 72% sulphuric acid. Following the extraction, the cellulose content was determined by the Seifert method described in reference [13], and alpha-cellulose, beta-cellulose, and gamma-cellulose content according to TAPPI T 203 cm-99. The last chemical analysis was the water solubility of the raw material under the TAPPI Test method T207 om-93 and the 1% sodium hydroxide solubility, according to TAPPI T 212 om-88.

The raw materials were disintegrated into chips of 1 cm length before the nitrate-alkaline cooking. The nitrate-alkaline cooking occurred in lab conditions by utilizing 2l boiling flasks. Each flask comprised 50 g of dry air material, and the ratio of the chemical to raw material was 20:1. The cooking process started with 45 min of cooking under the reflux condenser in 6% nitric. The washing by water was followed by the extraction in 5% sodium hydroxide when we heated the solution and then cooked for 10 min. We did not perform the heating under the reflux condenser. The additional washing by water was followed by another pulping and the neutralization by 1% acetic acid for 5 min. 

Water thoroughly washed the cooked pulp after cooking, and the rejects were separated from the pulp. The handsheets were made from the nitrate-alkaline pulp in a handsheet forming machine RAPID-KOTHEN (Birkenau, Germany). The mechanical properties of handsheets were tested to determine tensile properties on a device by the FRANK-PTI company (Birkenau, Germany) according to the ISO 1924-2 standard [14], such as breaking length, tensile index, or tensile absorption index. The smoothness by Bekk was determined on the device by FRANK-PTI according to the ISO 5627 standard [15] so that the results can be reproduced.

## 3. Results

### 3.1. Chemical Analysis

The chemical composition of input material has a significant impact on pulp yield and fiber characteristics. Cellulose is the principal component of fibers. Cellulose is the non-cellulose component of the cell walls and includes hemicellulose, pectin, lignin, protein, and also certain minerals. Table 1 contains the average chemical properties of black mustard (*Brassica Negra* L.) and camelina (*Camelina Sativa* L.), and we measured four samples for each analysis. As is known, the chemical composition of plants depends on the plant part, plant genotype, climate, and even the growing locality. This may be the reason why some of the chemical composition values differ from similar studies.

Table 2 shows the solubility of cellulose. The properties were assessed by solubility in alkali and alpha-cellulose from a practical point of view. Alpha-cellulose is the part that is nonsoluble in 17.5% sodium hydroxide. In contrast, after acidification, we excluded beta-cellulose, which is not dissolved in sodium hydroxide. The gamma-cellulose remains in the solution even after acidification. From the chemical point of view, the alpha-cellulose is cellulose in the meaning of its chemical definition but comprises a small number of hemicelluloses. The beta-cellulose originally does not exist in wood and plants. It arises from the destruction of cellulose. The gamma-cellulose comprises hemicelluloses [16].

Besides, the chemical analysis of raw materials is the determination of solutions extractible to organic and polar solvents. At first, a ball mill disintegrated the sample. The solvents for the experiments were acetone, a binary mixture of ethanol-toluene, cold water, hot water, and 1% solution of sodium hydroxide. Table 3 contains the concentrations of solutions which were extracted.

### 3.2. Nitrate-Alkaline Pulping

Nitrate-alkaline delignification was processed for the black mustard (*Brassica Negra* L.) and camelina (*Camelina Sativa* L.). The ratio of chemicals was 20:1 both for cooking in 6% nitric acid and for extraction in 5% sodium hydroxide.

The delignification degree for the cooked pulp was determined by the Kappa number, which was 20.2 for the pulp from black mustard and 23.1 for the camelina. According to these values, the cooking with nitric acid and the extraction in sodium hydroxide reaches the Kappa number, which is common for long-cook pulps or the pulps that were not bleached. Therefore, we can claim that significant delignification occurs during this process. The final yield was not as small as should be expected for a long cook time or after the bleaching process. The pulp processed this way was used for paper handsheets, making for further analysis as a tensile index. Table 4 comprises data that closely describe the general properties and value of smoothness by Bekk for the reproduction of the pulps from these pulps.

### 3.3. Pulp Characteristics

Table 5 contains the mechanical properties of nitrate-alkaline paper for plants measured by us compared with authors who examine other agricultural plants. The measure occurred for ten samples, and Table 5 shows the average values.

It is obvious from Table 5 that black mustard has better characteristics than camelina. The black mustard also has outstanding characteristics according to the comparison of the nitrate-alkaline paper characteristics from black mustard with papers made from the other non-wood plants processed by other methods. Only the sugarcane ethanol-soda pulp has higher strength values than the black mustard paper. The authors [20] achieved values for okra stalks soda pulp. With non-wood papers, we compared the measured values within industrially made papers, such as sulfite paper, kraft paper, and parchment paper. Figure 2 describes this comparison.

## 4. Discussion

The Seifert method, which was chosen to determine the cellulose, provides an almost identical content of cellulose as chromatography, which is the most accurate [13]. The content of cellulose by Seifert was similar for both plants. Comparing the rapeseed straw (33.9%) [6] to the corn (33.6%) [22] and the amaranth (31.9%) [5], the values were lower than the values measured by us. In contrast, the values of the sunflower (37.5%) [22] and wheat (38.2%) [6] were similar. The values were lower compared to the wood values. Barbash et al. [8] stated the values for birch (41.0%) and pine (47.0%).

Klason’s method analyzed the lignin, and the quantity of lignin and camelina is very similar to deciduous compared to wood (21.4% of lignin for oak and 24.5% for beech [23]).

In contrast, the Klason lignin value for black mustard is approximate to values of conifers; 29.5% for pine and 30.4% for spruce. Housseinpour et al. [22] stated the values for wheat (15.3%), rapeseed (20.0%), corn (17.4%) and sunflower (18.2%). These values are very similar to the amount of lignin measured by us.

The ash amount is similar for both plants, lower compared to the other non-wood plants. Barbash et al. [8] stated the amount of ash was 4.5% for wheat and rapeseed and Carvalho et al. [19] measured 2.3% for bagasse. Housseinpour et al. [22] analyzed the values for sunflower and corn, which were significantly higher, sunflower ash was 8.2%, and corn ash was 7.5%, which can be caused by the larger pith in stems. The amount of ash in the analysis of the trees was significantly lower than the values measured for non-wood plants, 0.5% for birch, and only 0.2% for pine [8].

The amount of silicates differs. Camelina contains twice as much more than black mustard. Compared to rapeseed, 0.0064% is significantly higher [7]. The values are higher compared to wood, which may be because we used the whole plant with fruits in our case. The paper [24] presents the values for pine (0.04%), and for beech and spruce (0.01%), which is higher than the values of rapeseed but significantly lower than the values for camelina and black mustard.

The alpha-cellulose values are significantly lower than for other plants; 28.8% for rapeseed [7]. The differences for other plants are even more substantial, 38.2% for wheat, 51.90% for bagasse, 48.8% for cotton stems, 41.2% and for rice [6]. Other authors, unfortunately, did not state the amount of the alpha-cellulose for other plants, nor did they state the amount of beta-cellulose and gamma-cellulose.

The amount of extractable solutions soluble in cold water is important for paper production, mainly in the process of wet pulping. Compared to the other authors, the solubility for camelina is lower than for the other field plants, but black mustard has similar solubility to wheat (10.7%) and rice (10.7%) [22]. On the contrary, the values for the other plants were higher, 13.8% for rapeseed, and the value for the sunflower was close to twice as high as camelina (16.5%) [22].

One would expect that the solubility in hot water is higher than in cold water because the polysaccharides of low molecular mass, which are nonsoluble in low temperature, can transition into the hot extract. The lower result corresponds to the fact that the mixed polysaccharides, polyuronic acids, and polyuronides are leaching into cold water and alkaline solution. These easily hydrolyze and already dissolve in cold water. Barbash et al. [8] stated the values for rapeseed (10.1%), wheat (6.0%), and pine (6.7%) which are values similar to those measured by us. On the contrary, he stated the value for birch only (2.2%). The plants that have larger pith, such as sunflower (15.5%), rice (16.2%), and corn (14.8%) [22], have a significantly larger amount of these extractives. It could cause a larger part of salts or saccharides in particular plants.

The number of plants measured by us in 1% of sodium hydroxide was 29.75% in the case of black mustard and 36.97% in the case of camelina. These values are similar to the ones for rapeseed stems (30.8%) [7] and sunflower (29.8%) [22]. However, the values for wood referred by Barbash et al. [8] are significantly lower, 11.2% for the birch and 19.4% for pine. The composition and the amount of extractives changes based on the species and age of the plant, even the location where the plant is grown and its parts. The hemicelluloses of non-wood plants mainly comprise pentosans, which can be isolated by extraction in diluted alkaline.

This may be the reason the amount of extractable solutions in 1% sodium hydroxide is higher than in other solvents. We can also assume that even the lignin is partly dissolved in the solution of sodium hydroxide.

The values of solubility in acetone are similar to the case of rapeseed stems 2.6% [7], cotton stems (2.9%) [6], bagasse (3.4%) [6], or of rice (3.5%) [22]. Low values can be caused because the mentioned plants do not contain such amount of the resin acids, which are soluble in acetone.

The transitional amount of dissolved solution in the binary ethanol-toluene mixture was only compared to the findings by Potůček et al. [7]. The stated values for rapeseed 5.3% are similar to camelina. The other authors focused on the chemical analysis do not state the solubility values of this mixture.

Since the aim of this study was the production of nitrate-alkaline pulp for non-bleached paper making or the utilization of pulp as the substitute for regenerated brown material using crop residues. The handsheets were made from nitrite-alkaline pulp from black mustard and camelina and were compared to commonly available papers according to their mechanical properties.

We achieved better properties for the pulp made from black mustard. Even in the case of the breaking length, we achieved 7.16 km, which came out better than for sulfite wood paper. Compared to the other authors, the breaking length, in our case of black mustard, was higher than soda pulp from rapeseed [7], soda pulp from corn [6], or chemical pulp made by peracetic acid from wheat [8]. In comparison with the kraft pulp (11.15 km), our pulp did not come out with such properties.

The other determining value was the relative elongation or so-called tensile stretch. That was, in our case, similar to the soda pulp from rapeseed [7]. Unfortunately, the other authors who pursued the pulp production from non-wood plants do not state this value.

In the case of the tensile index, the value for nitrate-alkaline pulp from black mustard was significantly lower than the values for soda pulp from okra [17] or the pulp produced by the ethanol-soda method from bagasse [19]. Compared to industrial pulps, it achieved lower values than kraft non-calender paper (59.42 Nmg^−1^) or parchment paper (52.46 Nmg^−1^), while the sulfite wood paper had even lowed values (20.68 Nmg^−1^). The last determining value was the tensile absorption index, which was compared only with the values of rapeseed cooked by the soda method [7], and these values were lower for the black mustard but higher than in the case of the camelina pulp. However, the mechanical properties of the non-wood pulp are of lower quality than the kraft pulp, but even the non-wood pulp has its benefits. Particularly the nitrate-alkaline cooking method because the black liquor shall fertilizer contains a large number of nitrates.

The N-supply plays a vital role in plant development, metabolism, and responses to stress. The authors of [25] could also conclude that N- application is useful for improving the overall growth, anatomical characteristics, and grain yield of wheat crops.

The European pulp and paper industry produces 11 million tons of waste a year [26]. The solution of composting of waste materials in paper production comprises discharging waste; respectively, the black liquor settles until most of the paper fibers and organic materials are stabilized (odor/chemically) through exposure to microorganisms with minimal carbon loss. Sometimes fertilizers are added to waste or black liquor to increase the nutrient content. This creates a humus material that can be used for application to agricultural land, houseplants, and greenhouses. This is one of the lowest disposal cost by-products in the paper industry. Except for the requirement of large land to spread the black liquor, there are several other composting costs [27,28,29,30]. Commercial composts must meet several technical requirements, such as maturity or suitability for plant growth. Composts made from organic waste mixed with various amounts of recycled paper and waste from the paper industry meet these requirements. The additional waste from the paper industry positively affected some parameters of compost, such as salt and organic content, and process into the composting matrix. The concentrations of harmful substances, heavy metals, in particular, should be considered as a limiting factor [31].

The application of composted MSW to agricultural land has several beneficial effects. Compost increases soil fertility by adding nutrients such as N, P, and K, thus substituting mineral fertilizers [32,33]. The addition of compost also increases plant health primarily by protecting against plant pathogens [34,35]. The addition of organic materials and the associated increase in soil organic matter (SOM) was associated with many positive effects such as improved soil structure, increased water holding capacity and infiltration, increased workability, and reduced erosion [36,37].

Several already issued and other foreseen European Union directives have a significant influence on the waste management strategy of paper-producing companies. The thermal processes, gasification, and pyrolysis seem to interest emerging options, although it is still necessary to improve the technologies for black liquor application. Other applications, such as hydrolysis to obtain ethanol, have several advantages (use of wet sludge and applicable technology to sludges), but these are not well developed for black liquor from the paper industry. Therefore, at this moment, the minimization of waste generation still has the highest priority.

## 5. Conclusions

Non-wood materials are the important raw material in countries where wood for paper and paper products production is not available in sufficient quantity. However, along with the growing consumption of paper and cellulose products, the quantity of wood sources is reducing, and the usage of non-wood raw materials for pulp production is growing [38]. Among the non-wood plants, the advantages are the annual renewal and the low costs of their renewal compared to the wood [9].

In the case of non-wood plants, the morphological and chemical composition differs. In the case of the low content of lignin in non-wood plants, there is a need for less energy and chemicals in bleached pulp production than in wood pulp production [39,40].

The non-wood plants can be used for paper production, but for now only by the soda cooking method, which represents an alkaline pulp production method without sulfur and during which the raw material is delignified in the sodium hydroxide solution in relatively drastic conditions while alkaline degradation of polysaccharides occurs. To avoid the alkaline degradation of polysaccharides, anthraquinone, ethylenediamine, oxygen, or methanol are added to the solution [41].

The summarized knowledge in this paper describes that non-wood plants are usable for pulp production, unfortunately only by the soda cooking method so far. However, the produced pulp mechanical properties are below the values for the bleached pulp made from wood by, as yet, the most used kraft method.

The chemical characterization of sugarcane trash showed that it contains a good amount of carbohydrates showing potential as a raw material for the pulp and paper industry, which is similar to our case [42].

Even though the crop fields in the Czech Republic are mostly sown under corn or rapeseed, this article was devoted to other plants that remain in the field, and their sown areas have been growing in the last few years.

With the resource utilization and efficient biological treatment of lignocellulosic biomass, the paper and pulp industry will be a model for sustainable development [43].

In future studies, it would be good to look at plants that are cooked by the industrial soda method to make the comparison of plants more concise. In addition, it would be good to discover the effect of black liquor after cooking the pulp on the amount of nitrogen in the soil.

## Figures and Tables

**Figure 1 materials-14-03673-f001:**
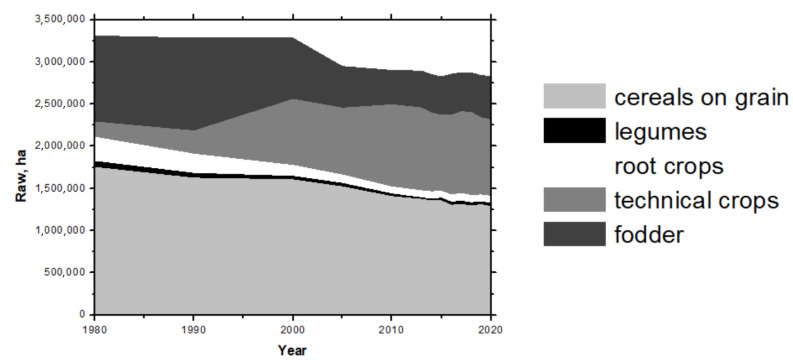
**Planting areas in the Czech Republic.** Cereals from grain (wheat, rye, barley, oat, and corn), legumes, root crops (potatoes, sugar beet), technical crops (rapeseed, sunflower, poppy, mustard, linen, etc.), and fodder [3].

**Figure 2 materials-14-03673-f002:**
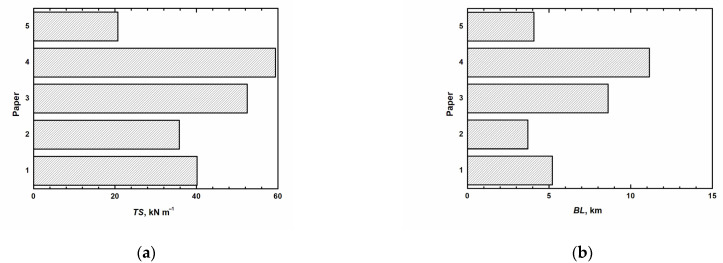
**Mechanical properties:** (**a**) Tensile index in N m g^−1^; (**b**) Breaking length in km (1-Nitrate-alkaline paper from black mustard, 2-Nitrate-alkaline paper from camelina, 3-Parchment paper, 4-Kraft paper, 5-Sulfite paper).

**Table 1 materials-14-03673-t001:** Chemical compounds (in mass %).

Raw	Cellulose	Lignin	Ash	Silica
Black Mustard (*Brassica Nigra* L.)	37.63	28.36	3.327	0.1058
Camelina (*Camelina Sativa* L.)	39.19	20.43	3.286	0.1945

**Table 2 materials-14-03673-t002:** Cellulose solubility (in mass %).

Raw	Alpha-Cellulose	Beta-Cellulose	Gamma-Cellulose
Black Mustard (*Brassica Nigra* L.)	15.09	9.38	3.02
Camelina (*Camelina Sativa* L.)	20.89	14.24	4.13

**Table 3 materials-14-03673-t003:** Extractives (in mass %).

Raw	Cold Water	Hot Water	1% NaOH	Acetone	Ethanol- Toluene
Black Mustard (*Brassica Nigra* L.)	10.49	8.33	29.75	2.23	10.15
Camelina (*Camelina Sativa* L.)	8.41	6.41	36.97	2.16	4.35

**Table 4 materials-14-03673-t004:** General properties of paper from the nitrate-alkaline pulp.

Raw	Kappa Number	Yield, %	Basis Weight, g m^−2^	Thickness, mm	Smoothness, s
Black Mustard (*Brassica Nigra* L.)	20.2	45.34	77.0	0.0955	21.25
Camelina (*Camelina Sativa* L.)	23.1	38.35	78.5	0.1265	15.53

**Table 5 materials-14-03673-t005:** Mechanical properties.

Paper	Raw	Breaking Length, km	Relative Elongation, %	Tensile Index, N m g^−1^	Tensile Absorption Index, J g^−1^
Nitrate-alkaline paper	Black Mustard	7.16	2.31	40.16	1.15
	Camelina	3.66	1.12	35.83	0.27
Soda paper	Rapeseed stalks [7]	4.06	1.70	–	0.47
	Rice straw [6]	–	–	26.10	–
	Corn [6]	0.30	–	3.20	–
	Okra stalks [17]	–	–	71.94	–
Neutral sulfite semi-chemical paper	Rapeseed residue [10]	3.63	–	35.60	–
Chemo-mechanical paper	Rapeseed straw [18]	–	–	37.70	–
	Bagasse [18]	–	–	33.50	–
Peracetic acid paper	Rapeseed straw [6]	4.20	–	–	–
	Wheat straw [6]	5.90	–	–	–
Ethanol-soda paper	Sugarcane bagasse [19]	–	–	80.00	–
Parchment paper	Mixture of sisal, hemp, and linen	8.62	1.42	52.46	–
Kraft paper	Softwood	11.15	5.62	59.42	–
	Miscanthus [20]	–	2.00	70.00	–
Sulfite paper	Softwood	4.07	1.45	20.68	–
	Softwood [21]	–	–	25.96	–

## Data Availability

Data are available on request due to ethical restrictions. The data presented in this study are available on request from the corresponding author. The data are not publicly available due to [unfinished research].

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
