# Peer review of "Nitrate-Alkaline Pulp from Non-Wood Plants"

_materials, 2021, doi:10.3390/ma14133673_

Round 1
Reviewer 1 Report
The manuscript under appreciation is about the suitability of Black Mustard and Camelina for pulp manufacturing and production of laboratory sheets.
The article is generally badly written. The aim of this study is not clearly presented. The experimental methods are not described appropriately. The chemical analysis results have not been conducted in triplicates and the results are unreliable. The presence of Figure 1 in the Results is inappropriate as the authors did not conduct any relevant research for the planting areas, therefore this information must have been acquired from a reference that the authors have not reported in the manuscript. The Conclusion section is chaotic, looks like the Discussion section.
Author Response
Reviewer 1:
In the attachment.
Dear Reviewer no. 1,
Thank you very much for reviewing our paper and providing us with additional recommendations.
We have tried to follow all of them to improve the quality of our paper.
The manuscript is focused on an important research topic, namely investigation and evaluation of the availability of lignocellulosic raw materials and their suitability for the production of pulp and paper, based on their chemical and mechanical properties.
You see that based on your recommendations. We have described the aim of the work better and presented experimental measurements in more detail. It is clear from the detailed description that the data are average, not that only one experiment was performed.
We made it easier for readers to keep knowledge of the planting areas in the manuscript–Figure 1. We created Figure 1 in a new form, with a proper citation added.
We also corrected the discussion and conclusion to your recommendation. The conclusion should be more concise and concise now.
With regards,
The Authors

Reviewer 2 Report
1-Abstract: add the results of work
2- The introduction should be extended to discuss the hypothesis and research questions in details. Additionally, the introduction should cover the recent literature related to this subject.
3- Material and methods: How many replicates per sample were performed for measurement of each property? Add to the text.
Some of the standards are not explain in a "References" part. Missing literature designation.
4- Results: Figure 1.: is not a good quality
line 101: not hemicellulose but hemicelluloses
The literature is not well numbered (line 117, literature n. 19)
line 138: How many sheets were prepared?
Is it the average value of Basis weight, thickness and smoothness? Please explain it in Table 4.
5-Discussion: good described
6-Conclussion: to extensive, missed the results of work
Výsledky prekladov
Author Response
Reviewer 2:
Dear Reviewer no. 2,
Thank you very much for reviewing our paper and providing us with additional recommendations.
We have tried to follow all of them to improve the quality of our paper.
With regards,
The Authors
1–Abstract: Add the results of work.
Thank you for your recommendation. We have added aims for the work.
2–The introduction should be extended to discuss the hypothesis and research questions in details. Additionally, the introduction should cover the recent literature related to this subject.
Thank you for your recommendation. We have improved that accordingly.
3–Material and methods: How many replicates per sample were performed for measurement of each property? Add to the next.
Thank you for your recommendation. We measured each sample four times, the average result is given. We have clarified accordingly.
Some of the standards are not explain in a “References” part. Missing literature designation.
Thank you for your recommendation. We have improved that accordingly.
4–Results: Figure 1.: is not a good quality
Thank you for your recommendation. We have improved that accordingly.
Line 101: not hemicellulose but hemicelluloses
Thank you for your recommendation. We have improved that accordingly.
The literature is not well numbered (line 117, literature n. 19)
Thank you for your recommendation. We have improved that accordingly.
Line 138: How many sheets were prepared?
Thank you for your recommendation. We produced ten sheets from both types of pulp for adequate measurement results. We have clarified that accordingly.
Is it the average value of Basis weight, thickness and smoothness? Please explain it in Table 4.
Thank you for recommendation. We have clarified that accordingly.
5 – Discussion: good described
Thank you.
6 – Conclusion: to extensive, missed the results of work
Thank you for your recommendation. We have slightly improved the conclusion.
Reviewer 3 Report
The authors describe a method to extract pulp for paper from non-wood plants found in the Czech Republic. Overall this is a useful work but there are several aspects of the work that need to be rewritten before it can be published.
- It is not clear from the abstract or the introduction what the work is really about and it's significance. Actually, the conclusion is a much better overview of the work than either the abstract or the introduction, but one shouldn't have to read to the end of the paper before properly understanding what it is about. The abstract and introduction need significant rewriting and the conclusion needs to be much more concise and only present the highlights of the work.
- There is actually no mention of the significance of the findings in the conclusion, which certainly needs to be addressed.
- Figure 1 is of very poor quality and needs to be improved
- There is no real comparison to paper from wood, which I would have expected as it is the standard for paper making. All the discussion compares to non-wood plants (which is certainly useful) but a comparison to standard paper is essential to understand the usefulness of the pulp.
- There is a lot of presentation of figures, quantities of different components of the plant make-up but no discussion of what this means in terms of usability, this also needs to be addressed.
- On line 264, there are two words of a sentence which does not continue
- A thorough grammar and spelling check must be done before publication
Author Response
Reviewer 3:
Dear Reviewer no. 3,
Thank you very much for reviewing our paper and providing us with additional recommendations.
We have tried to follow all of them to improve the quality of our paper.
With regards,
The Authors
The authors describe a method to extract pulp for paper from non-wood plants found in the Czech Republic. Overall this is a useful work but there are several aspects of the work that need to be rewritten before it can be published.
- It is not clear from the abstract or the introduction what the work is really about and it's significance. Actually, the conclusion is a much better overview of the work than either the abstract or the introduction, but one shouldn't have to read to the end of the paper before properly understanding what it is about. The abstract and introduction need significant rewriting and the conclusion needs to be much more concise and only present the highlights of the work.
Thank you for your recommendation. We have improved accordingly and shortened the conclusion.
- There is actually no mention of the significance of the findings in the conclusion, which certainly needs to be addressed.
Thank you for your recommendation. We have clarified that accordingly. - Figure 1 is of very poor quality and needs to be improved
Thank you for your recommendation. We have improved Figure 1 accordingly and improved their quality.
- There is no real comparison to paper from wood, which I would have expected as it is the standard for paper making. All the discussion compares to non-wood plants (which is certainly useful) but a comparison to standard paper is essential to understand the usefulness of the pulp.
Thank you for your recommendation. We have clarified in Table 5 accordingly.
- There is a lot of presentation of figures, quantities of different components of the plant make-up but no discussion of what this means in terms of usability, this also needs to be addressed.
Thank you for your recommendation. We have improved the manuscript and provided more elaborated discussion.
- On line 264, there are two words of a sentence which does not continue.
Thank you for your recommendation. We have clarified that accordingly.
- A thorough grammar and spelling check must be done before publication
Thank you for your recommendation. We have corrected that accordingly.
Thank you again.
Round 2
Reviewer 1 Report
The authors have addressed the majority of the issues and the manuscript has been significantly improved. However, I strongly recommend checking the English presentation throughout the manuscript with an emphasis on the syntax and the use of passive voice.
For example:
line 67: On the fields in the highlands of Ore mountains grown the black mustard (Brassica Nigra L.), whereas in the lowlands of Podyjí area harvested the camelina (Camelina Sativa L.).
The samples of the black mustard (Brassica Nigra L.) were originated from the highlands of the Ore mountains, whereas the camelina (Camelina Sativa L.) samples were harvested in the lowlands of Podyjí area.
line 71: We chemical determined analysis for randomly taken samples in a laboratory vibratory mill....
The chemical analysis of the randomly taken samples was performed using a laboratory vibratory mill....
Author Response
Reviewer 1:
In the attachment.
Dear Reviewer no. 1,
Thank you very much for reviewing our paper and providing us with additional recommendations.
We have tried to follow all of them to improve the quality of our paper.
With regards,
The Authors

Reviewer 3 Report
After re-reading this paper, there is not significant improvement to warrant publication at this stage. It is still not clear from the abstract and introduction, why the authors specifically choose the plants they have chosen and what contribution this will make. Additionally, the English must be edited significantly to make the paper clear to the reader. Please consider these aspects before choosing to re-submit.
Author Response
Reviewer 3:
In the attachment.
Dear Reviewer no. 3,
Thank you very much for reviewing our paper and providing us with additional recommendations.
We have tried to follow all of them to improve the quality of our paper.
With regards,
The Authors
